# Survival Outcomes and Failure Patterns in Patients with Inoperable Non-Metastatic Pancreatic Cancer Treated with Definitive Radiotherapy

**DOI:** 10.3390/cancers15082213

**Published:** 2023-04-09

**Authors:** Biyang Cao, Letian Zhang, Chenchen Wu, Xiaoliang Liu, Qianqian Wang, Fang Tong, Wei Yang, Jing Wang

**Affiliations:** 1Chinese PLA Medical School, Beijing 100853, China; 2Department of Radiation Oncology, The First Medical Center, Chinese PLA General Hospital, Beijing 100853, China

**Keywords:** pancreatic cancer, definitive radiotherapy, long-term survival, failure patterns, prognostic factor

## Abstract

**Simple Summary:**

Historically, the only curative option for patients with pancreatic cancer was surgical resection. It is essential to develop nonoperative local treatment options that can provide a similar benefit. With the development of updated radiotherapy techniques that address organ motion, patients with inoperable diseases have been able to receive curative radiation doses. However, based on modern radiotherapy techniques, the survival outcomes and failure patterns of definitive radiotherapy to the primary tumor for inoperable pancreatic cancer remain inadequately reported in a large population. This study was performed to elucidate the effectiveness of contemporary treatment, identify key failure patterns, and determine the late toxicity in patients treated with definitive radiotherapy, according to our institutional data from the last five years.

**Abstract:**

This study investigated the long-term results, failure patterns, and prognostic factors of patients with initially inoperable non-metastatic pancreatic cancer (PC) receiving definitive radiotherapy (RT). Between January 2016 and December 2020, a total of 168 non-metastatic PC patients, who were surgically unresectable or medically inoperable, were enrolled to receive definitive RT, with or without chemotherapy. Overall survival (OS) and progression-free survival (PFS) were evaluated using the Kaplan–Meier method with a log-rank test. The cumulative incidence of locoregional and distant progression was estimated using the competing risks model. The Cox proportional-hazards model was used to determine the influence of prognostic variables on OS. With a median follow-up of 20.2 months, the median OS (mOS) and median PFS (mPFS) from diagnosis were 18.0 months [95% confidence interval (CI), 16.5–21.7 months] and 12.3 months (95% CI, 10.2–14.3 months), respectively. The mOS and mPFS from RT were 14.3 months (95% CI, 12.7–18.3 months) and 7.7 months (95% CI, 5.5–12.0 months), respectively. The corresponding 1-year, 2-year, and 3-year OS from diagnosis and RT were 72.1%, 36.6%, and 21.5% as well as 59.0%, 28.8%, and 19.0%, respectively. In a multivariate analysis, stage I–II (*p* = 0.032), pre-RT CA19–9 ≤ 130 U/mL (*p* = 0.011), receiving chemotherapy (*p* = 0.003), and a biologically effective dose (BED_10_) > 80 Gy (*p* = 0.014) showed a significant favorable influence on OS. Among the 59 available patients with definite progression sites, the recurrences of local, regional, and distant progression were 33.9% (20/59), 18.6% (11/59), and 59.3% (35/59), respectively. The 1-year and 2-year cumulative incidences of locoregional progression after RT were 19.5% (95% CI, 11.5–27.5%) and 32.8% (95% CI, 20.8–44.8%), respectively. Definitive RT was associated with long-term primary tumor control, resulting in superior survival in patients with inoperable non-metastatic PC. Further prospective randomized trials are warranted to validate our results in these patients.

## 1. Introduction

The rate of incidence and deaths from pancreatic cancer (PC) are increasing worldwide. It is estimated that PC will become the second-leading cause of cancer-related death in the next ten years [1]. According to the reports from National Cancer Institute (NCI), the five-year survival rate for PC is only 11.5% [2]. Surgery is the sole potentially curative option for PC. However, most patients suffer from locally advanced or metastatic disease, where radical surgery cannot be performed [3]. Furthermore, in those patients with operable disease, a notable proportion of patients, especially in older subsets, are medically inoperable or unwilling to undergo radical surgery, through consideration of the significant comorbidities or diminished performance status [4].

The optimal treatment for these cases that are surgically unresectable or medically inoperable remains uncertain, though current treatment often includes systematic chemotherapy in combination or not with radiotherapy (RT). Unfortunately, patients suffering from unresectable disease are incurable. Therefore, chemotherapy is used to alleviate the progression of the disease and prolong life [5]. RT emphasizes local tumor control, such as primary tumor, as well as relatively isolated and localized foci. Although distant progression is the leading cause of death in PC as a whole, up to one third of patients experience local progression, which may negatively impact their quality of life or survival [6,7]. Local tumor progression may cause pain, gastrointestinal obstruction, or bleeding, which can significantly impact a patient’s quality of life. Thus, the control of local tumors remains a crucial aspect of treatment of PC patients who are not candidates for curative surgery.

Non-operative local therapeutic options that can provide similar benefits are required. Contemporary radiotherapy can delivery ablative radiation doses with engraving of target areas and precisely tracking them, to minimize the danger for organs at risk [8]. Emerging radiation techniques, such as image-guided (IG), intensity-modulated (IM), and dose-guide (DG) radiation therapies; volumetric-modulated arc therapy (VMAT) and helical tomotherapy; and stereotactic body radiation therapy (SBRT), including Cyberknife, have made this possible. Studies based on modern RT techniques in small cohorts suggest that primary tumor control may enhance survival when doses are escalated to an ablative threshold [9,10,11]. However, details of long-term results and patterns of failure for definitive RT to the primary tumor in inoperable PC patients remain inadequate. In this study, we reviewed our institutional experience, to evaluate the efficacy of contemporary definitive RT, determine the most common failure patterns, and assess the toxicity in patients with inoperable non-metastatic PC.

## 2. Materials and Methods

### 2.1. Patient Selection

Between January 2016 and December 2020, one hundred sixty-eight inoperable non-metastatic PC patients who had received definitive RT with chemotherapy or without in the Department of Radiation Oncology of Chinese PLA General Hospital, Beijing, China, were enrolled in our study. These patients were surgically unresectable or medically inoperable, such as those with potentially operable disease with a comorbidity that prevented surgery (stage I–II) and locally advanced unresectable cancer (stage III), as per the 8th edition of the American Joint Committee on Cancer (AJCC) staging system. The determination of PC diagnosis was based on clinical, radiological, and pathological findings discussed in a multidisciplinary consult team, in accordance with the current guidelines. The inclusion criteria were as follows: (1) aged 18 years or more; (2) clinically or pathologically confirmed diagnosis of PC between 2016 and 2020; (3) no surgery performed; (4) no distant organ metastasis at diagnosis; (5) intent to treat with definitive RT for primary tumor; (6) Eastern Cooperative Oncology Group (ECOG) performance status score less than or equal to 2; and (7) essentially normal major organ and bone marrow function. The exclusion criteria included: (1) patients with recurrence or metastasis after pancreatectomy; (2) history of previous or concurrent malignancies at other sites; (3) pancreatic neuroendocrine tumor; (4) a history of previous upper abdominal RT; (5) contraindications to RT; (6) uncontrollable comorbidities; and (7) missing follow-up. The detailed patient selection process is shown in Figure 1.

The study protocol was in accordance with the Declaration of Helsinki ethical guidelines and was endorsed by the independent ethics committees at The First Medical Center of Chinese PLA General Hospital. Patient consent was waived, given the retrospective nature of the study.

### 2.2. Treatment

#### 2.2.1. IG-IMRT

Patients were treated with IG-IMRT, with dose escalation through simultaneous integrated boost, which was administered using either Tomotherapy^®^ (Accuray^®^ Incorporated, Sunnyvale, CA, USA) or RapidArc^®^ (Varian Medical Systems, Palo Alto, Santa Clara, CA, USA) treatment planning systems. All patients were subjected to body plate fixation, contrast, and kilovolt (KV) computed tomography simulation. Fused computed tomography contrast images were used to contour the gross target volume (GTV). Enhanced magnetic resonance (MRI) or positron emission tomography (PET) images were utilized to help determine the contours of the target. The GTV was identified as the pancreatic primary lesions and metastatic lymph nodes. The clinical target volume (CTV) was determined in accordance with GTV and the suspected involvement field or drainage area. PTV comprised GTV as well as CTV with a margin of 5–6 mm in the axial plane and 5–10 mm for the longitudinal axis without protruding into the intestinal tract and the stomach [12]. An adjustment was made to the margin based on the location of the tumor and the gastrointestinal tract. The dose of PTV/GTV was 50 Gy/60–65 Gy in 25–30 fractions. Daily image-based guidance was implemented, utilizing cone beam computed tomography imaging to verify the setup before each faction. Dose–volume limitations for organs at risk (OAR) were based on the NCCN guidelines on conventional fractionation RT.

#### 2.2.2. SBRT

The delivery of SBRT was via Cyberknife (Accuray Incorporated, Sunnyvale, CA, USA) in our study. One to three gold fiducials were implanted in the lesion under the guidance of abdomen ultrasound or computed tomography before SBRT. The delineation of GTV and OAR was contoured using plain and contrast-enhanced parenchymal computed tomography. MRI or PET was recommended, to aid in target definition. The contour of GTV included primary tumor and metastatic lymph nodes. The PTV included 0–5 mm enlargement of GTV. The contour of the PTV cannot stretch into and overlap the margin of the gastrointestinal tract. The prescribed dose of PTV was 42.5–55.0 Gy, in 5–7 fractions. Respiration synchronous tracking (Synchrony) was utilized to monitor the fiducials movement in the course of simultaneous irradiation. ¡ Timmerman tables were used as a reference for dose constraints for OAR [13].

### 2.3. Chemotherapy

Our institutional protocol does not specify chemotherapy regimens prior to and after RT. Chemotherapy regimens referred to principles of system therapy for pancreatic adenocarcinoma from the NCCN guidelines [5], including FOLFIRNOX, Gemcitabine combination, Albumin-bound Paclitaxel combination, and Fluoropyrimidine combination and the single versions.

### 2.4. Follow-Up and Endpoints

Patients were followed up regularly after receiving RT. This included physical examination, assessment of toxicity related to treatment, radiographic examination of the abdomen and pelvis, and testing for the cancer antigen 19-9 (CA 19-9). Patients received follow-up examinations after 3, 6, and 12 months of treatment in the first and second years after diagnosis, and every six months thereafter. For all patients, the time of recurrence and the sites of progression were documented. The last follow-up visit was on 30 April 2022.

Primary endpoints included overall survival (OS) and progression-free survival (PFS). The secondary endpoint included the rate of local and distant progression. The overall survival was measured starting from the diagnosis date, as well as the initial fraction of RT to the time of death, evaluating patients at the end of the follow-up.

The definition of local and distant progression of disease was based on the response evaluation criteria in solid tumors. Radiation Therapy Oncology Group (RTOG) and European Organization For Research and Treatment Of Cancer (EORTC) criteria were used to define and evaluate the toxicity associated with treatment.

### 2.5. Statistical Analysis

Categorical variables are presented as frequencies in percentages, and Chi-square tests are used to compare them. The cutoff values of CA19–9 and BED_10_ were determined using X-tile software (Yale University School of Medicine, New Haven, CT, USA).

All patients with intention to treat were included for analysis of efficacy and adverse effects. The OS and PFS were estimated using the Kaplan–Meier method with a log-rank test. Univariate and multivariable analyses were performed using a Cox proportional hazards model. The cumulative incidence of local and distant progression was estimated using the Fine–Gray competing risks regression model. Venn diagrams were plotted to demonstrate the patterns of treatment failure. Statistical analyses were conducted using R software (version 4.0.1; http://www.r-project.org, accessed on 12 February 2023).

## 3. Results

### 3.1. Patient and Treatment Characteristics

A summary of the baseline characteristics of the cohort is presented in Table 1. The median age for all patients was 64 years (range, 36–85 years), with 69 females and 99 males. Tumors in the head, body, or tail of the pancreas accounted for 66.1% (111/168) and 33.9% (57/168), respectively. Fifty patients (29.8%) with technically operable disease (stage I–II) were not able to undergo surgery because of their comorbidities. One hundred eighteen patients (70.2%) had locally advanced unresectable disease (stage III) at diagnosis; 18.5% (31/168) and 81.5% (137/168) of patients were treated with IG-IMRT and SBRT, respectively. Overall, 95 patients were treated with RT alone, and 73 patients had RT combined with chemotherapy (combined modality treatment, CMT). Of the patients treated with CMT, 52 patients (71.2%, 52/73) received pre-RT chemotherapy and 47 patients (64.3%, 47/73) received post-RT chemotherapy. According to sequential patterns of RT and chemotherapy, 26 patients (35.6%) received induction chemotherapy followed by radiotherapy and consolidation chemotherapy (CT-RT-CT), 26 patients (35.6%) received chemotherapy followed by radiotherapy (CT-RT), and 21 patients (28.8%) received radiotherapy followed by chemotherapy (RT-CT). Among all patients, the median interval time from diagnosis to RT was 2.2 months (range 1.0–31.7 months), including 6.2 months for those receiving pre-RT chemotherapy and 1.6 months for those who did not.

### 3.2. Overall Survival

With a median follow-up time of 20.2 months [95% confidence interval (CI),17.8–33.1 months] measured from diagnosis, 91 patients (54.2%) were dead and 77 patients (45.8%) were alive. The median OS (mOS) and median PFS (mPFS) from the time of diagnosis were 18.0 months (95% CI, 16.5–21.7 months) and 12.3 months (95% CI, 10.2–14.3 months), respectively (Figure 2A,C). The corresponding 1-year and 2-year OS from the time of diagnosis were 72.1% (95% CI, 64.9–80.1%) and 36.6% (95% CI, 28.3–47.3%), respectively. The mOS was 21.1 months (95% CI, 17.0–46.0 months) for stage I–II patients and 17.0 months (95% CI, 16.0–22.2 months) for stage III patients.

### 3.3. Survival Outcomes According to Definitive RT

With a median follow-up time of 16.2 months (95% CI, 15.6–30.3 months) from the time of RT, the mOS and mPFS after RT were 14.3 months (95% CI, 12.7–18.3 months) and 7.7 months (95% CI, 5.5–12.0 months), respectively (Figure 2B,D). The corresponding 1-year and 2-year OS were 59.0% (95% CI, 49.4–67.2%) and 28.8% (95% CI, 28.1–39.4%), respectively. The mOS was 18.0 months (95% CI, 14.1–44.4 months) for stage I–II patients and 14.1 months (95% CI, 10.0–17.8 months) for stage III patients. According to the RT techniques, the mOS after RT for 31 patients treated with IMRT and 137 patients with SBRT were 12.7 months (95% CI, 8.8–27.9 months) and 14.6 months (95% CI, 13.3–20.1 months). The mOS after RT in the two groups of patients with BED_10_ ≤ 80 Gy (*n* = 54) and BED_10_ > 80 Gy (*n* = 114) were 8.8 months (95% CI, 5.9–17.5 months) and 15.7 months (95% CI, 14.1–23.2 months).

### 3.4. Survival Outcomes According to Treatment Modalities by Stage

The outcomes of the treatment modalities were analyzed in subgroups of AJCC stage. In patients with stage I–II disease, the mOS from diagnosis in the RT group (*n* = 37) and CMT group (*n* = 13) was 21.1 and 20.4 months (*p* = 0.47), respectively (Figure 3A). In patients with stage III disease, the mOS from diagnosis in the RT group (*n* = 58) and CMT group (*n* = 60) was 13.8 months and 19.2 months (*p* = 0.0044), respectively (Figure 3A). In patients with stage III disease treated with CMT, the mOS of the CT-RT-CT group (*n* = 23), CT-RT group (*n* = 23) and RT-CT group (*n* = 14) was 24.1, 17.0, and 18.0 months (*p* = 0.036), respectively (Figure 3B).

### 3.5. Prognostic Factors

Patient and treatment characteristics were assessed for their prognostic significance for OS from the time of diagnosis. Univariate analysis revealed that AJCC stage, pre-RT CA19–9 levels, chemotherapy, and BED_10_ were all significantly associated with OS from the time of diagnosis. Kaplan–Meier plots of OS by chemotherapy and BED_10_ are shown in the Figure 4. In the multivariable analysis, stage I–II (HR: 0.58, 95% CI: 0.35–0.95, *p* = 0.032), pre-RT CA19–9 ≤ 130 U/mL (HR: 0.57, 95% CI: 0.37–0.88, *p* = 0.011), receiving chemotherapy (HR: 0.51, 95% CI: 0.33–0.79, *p* = 0.003), and BED_10_ > 80 Gy (HR: 0.54, 95% CI: 0.33–0.88, *p* = 0.014) were identified as independent prognostic factors favorable for OS (Table 2).

### 3.6. Treatment-Related Toxicity

Except for seven patients who discontinued RT due to disease progression, all others completed RT without treatment-related deaths. Treatment-related toxicities are shown in Table 3. The most common acute toxicities included nausea and vomiting (79.2%, 133/168), neutropenia (36.9%, 62/168), thrombocytopenia (9.5%, 16/168), and gastrointestinal ulcer (1.2%, 2/168), with grade 3 or higher nausea and vomiting and hematologic toxicities accounting for 6.8% and 5%, respectively. Late toxicity included gastric ulcers (3.0%, 5/168), duodenal ulcers (8.3%, 14/168), and gastrointestinal obstruction (9.5%, 16/168), with grade 3 or higher gastrointestinal obstruction accounting for 2.4%.

### 3.7. Disease Progression and Failure Patterns

By the last follow-up, 54.2% of patients (91/168) had experienced disease progression. The 1-year and 2-year cumulative incidences of disease progression after RT in the entire group were 61.2% and 79.1% (Appendix A). Among the 59 available patients with definite progression sites, the 1-year and 2-year cumulative incidences of locoregional progression were 19.5% (95% CI, 11.5–27.5%) and 32.8% (95% CI, 20.8–44.8%) (Figure 5A), respectively. The 1-year and 2-year cumulative incidences of locoregional progression in the two BED groups (≤80 Gy vs. >80 Gy) were 29.7% vs. 16.1% and 52.3% vs. 28.1% (Figure 5B). The cumulative incidences of distant progression at the first and second year were 31.5% (95% CI, 21.8–41.1%) and 33.4% (95% CI, 23.3–43.6%), respectively (Figure 5C). The 1-year and 2-year cumulative incidences of distant progression in the two BED groups (≤80 Gy vs. >80 Gy) were 36.3% vs. 29.4% and 36.3% vs. 31.9%,respectively (Figure 5D). Figure 6 displays the patterns of disease failure in the 59 available patients with clear progression sites. The local failure, regional failure, and distant metastasis accounted for 33.9% (20/59), 18.6% (11/59), and 59.3% (35/59), respectively. The common sites of distant metastasis observed were liver (77.1%, 27/35), peritoneum and/or peritoneal cavity (11.4%, 4/35 patients), lung (11.4%, 4/35), and bone (8.6%, 3/35).

## 4. Discussion

The present study reported the survival outcomes and failure patterns in initially inoperable non-metastatic pancreatic cancer patients with intention for definitive RT, in a large volume based on real-world data. The observed 1-year cumulative incidence of locoregional progression was 19.5% from the start of definitive RT, which is comparable to other recent studies using ablative RT in patients with PC [14,15]. There was a significant positive correlation between the improvement in local progression-free survival and radiotherapy (especially BED_10_). The 1-year and 2-year cumulative incidences of locoregional progression in the BED group (≤80 Gy vs. >80 Gy) were 29.7% vs. 16.1% and 52.3% vs. 28.1%. These results suggest that local control with definitive RT has the potential to translate into a survival benefit. In the present study, the observed mOS for patients with stage I–III, stage I–II, and stage III were 18.0, 21.1, and 17.0 months, respectively. Compared to historical data with a mOS of 8.6 to 15.2 months in non-resected stage III patients who received conventional RT after induction gemcitabine [16], the present study showed a significant survival benefit. Compared to the report of a mOS of 17 to 38 months in stage I–II patients who underwent surgical resection in accordance with the AJCC 8th edition staging system [17], our results showed a favorable and encouraging survival outcome with definitive RT for a primary tumor.

Although curative surgical resection followed by adjuvant chemotherapy is the standard treatment for early-stage PC [18], up to 28% of patients had postoperative complication morbidities or mortality due to comorbidities [19]. Moreover, it is estimated that over 40% of patients with PC are diagnosed at an advanced age (≥75 years), many of whom have comorbid illnesses or poor performance status, which precludes them from receiving aggressive therapy [20]. The optimal treatment remains uncertain for patients who are “medically inoperable”. The NCCN guidelines recommend chemotherapy with a single agent and palliative RT or supportive care for poorly performing patients [5]. In the present study, subgroup analysis according to the treatment modality (RT alone vs. CMT) in patients with stage I–II indicated that the mOS were 21.1 months and 20.4 months, respectively. There were no significant differences observed between the two groups (*p* = 0.47). Given the demonstrated high efficacy of RT, the relative intolerance to chemotherapy, and the lack of benefit of adding chemotherapy to RT, definitive RT has the potential to be the primary treatment for patients with medically inoperable early-stage PC.

Approximately 30% of newly diagnosed cases are classified as locally advanced unre- sectable PC (LAUPC), because of the involvement of adjacent vital vessels. The administration of neoadjuvant chemotherapy or chemoradiotherapy (CRT) has brought the opportunity for curative resection. However, such a conversion is rare for LAUPC patients. Current guidelines for LAUPC recommend non-operative treatment using a multidisciplinary approach. The optimal therapy regimen and sequence for LAUPC patients remains unclear. Options for treatment included induction chemotherapy followed by CRT or SBRT, CRT followed by chemotherapy, or multi-agent chemotherapy or a single agent alone. There is controversy regarding the role of RT in the management of LAUPC, with five phase-3 randomized controlled trials showing conflicting results [21,22,23,24,25]. A contemporary LAP07 trial demonstrated an improvement in reducing local progression rate (32% vs. 46%; *p* = 0.03) in patients who received conventional CRT following four cycles of induction chemotherapy, but did not demonstrate a survival benefit (16.5 vs. 15.2 months; *p* = 0.08). Recent trials in patients with LAUPC, utilizing modern and more intensive chemotherapy regimens, demonstrated that RT is independently related with improved OS [26,27,28].

In the contemporary era, however, there is a growing emphasis on understanding the role of more active chemotherapy regimens and precise radiation techniques. Intensive systemic therapy for LAUPC helps to reduce distant metastasis, but local disease progression may limit long-term survival. Even in studies of individuals with PC who undergo surgical resection and adjuvant chemotherapy, which is deemed curative, local progression remains the predominate pattern of failure, indicating that all PC patients require intensive local therapy. The modern techniques including IMRT and SBRT with dose escalation have made precise radiotherapy available and have emerged as novel therapeutic options for PC care over the past decade. In this series, the combination of chemotherapy and RT remains the preferred therapeutic option for individuals with LAUPC (RT alone vs. CMT, mOS 13.8 months vs. 19.2 months, *p* = 0.0044). Furthermore, the effects of sequential patterns of RT and chemotherapy on survival were also explored in an analysis of patients treated with CMT. The mOS for the CT-RT-CT, CT-RT, and RT-CT groups were 24.1, 17.0, and 18.0 months, respectively (*p* = 0.036). The CT-RT-CT group had a significantly better survival outcome than the CT-RT and RT-CT groups. CT-RT-CT seems to be a preferred sequential paradigm for certain LAUPC patients. Currently, the NCCN guidelines recommend the use of an initial course of CT followed by CRT for LAUPC patients with stable disease or an objective response to the initial chemotherapy. This strategy aims to control systemic disease, while identifying patients with undetected metastatic disease who are unlikely to benefit from CRT.

There were several limitations identified in this study. First of all, it was a retrospective, non-randomized, and single-institution study, with possible selection bias. Second, a variety of chemotherapy regimens were administered to the patients prior to, or following RT. Our endpoints of interest may have been biased by the heterogeneity of the chemotherapy regimens used, since chemotherapy affects patients’ survival, progression, and toxicity outcomes. The current study enrolled patients with stage I–III disease, whereas previous studies only recruited patients with locally advanced disease. The inclusion of a diverse group of patients increased the generalizability of the study. However, including all stages of patients limits our ability to draw conclusions regarding the effects of RT on survival and local control. In comparison with the previous literature analyzing patients with stage III disease only, our study provides a more comprehensive analysis. According to our research, definitive RT is a promising treatment for patients with PC, resulting in better local control. Further research is needed to determine the optimal dose and the impact of RT on survival.

## 5. Conclusions

Definitive RT was associated with maintained control of the primary tumor, contributing to a favorable survival outcome with inoperable non-metastatic PC. Further prospective randomized trials are warranted to validate our results.

## Figures and Tables

**Figure 1 cancers-15-02213-f001:**
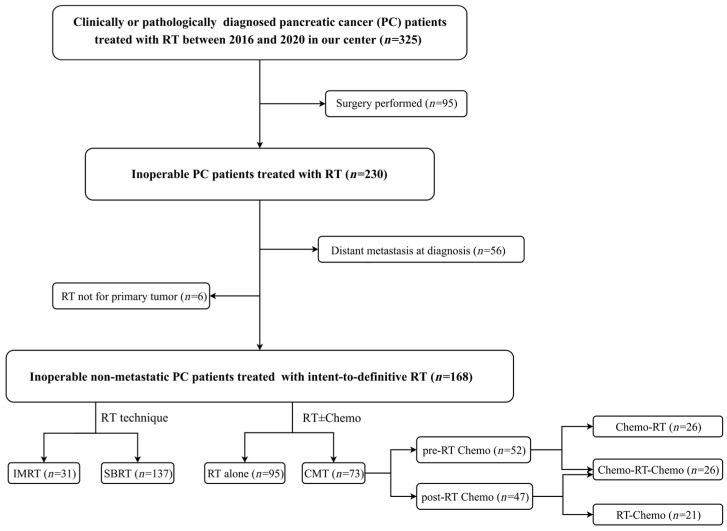
Flow chart of study enrollment and exclusions.

**Figure 2 cancers-15-02213-f002:**
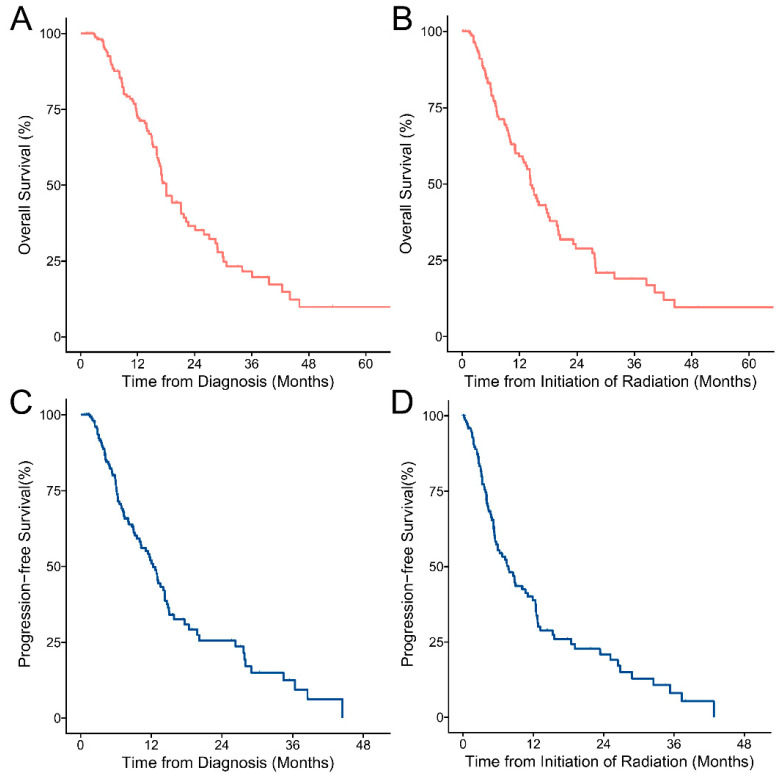
Kaplan–Meier plots of overall survival (OS) and progression-free survival (PFS) of the entire cohort. (**A**) OS measured from the date of diagnosis; (**B**) OS measured from radiotherapy; (**C**) PFS measured from diagnosis; (**D**) PFS measured from radiotherapy.

**Figure 3 cancers-15-02213-f003:**
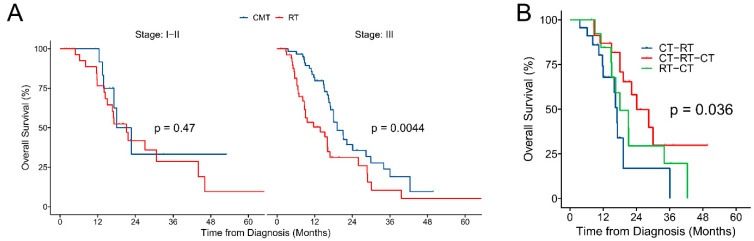
Survival analysis according to treatment modalities by stage. (**A**) Overall survival (OS) from diagnosis with radiotherapy alone (RT) or combined modality therapy (CMT) in patients with stage I–II and III disease; (**B**) OS from diagnosis with different sequence of radiotherapy (RT) and chemotherapy (CT) in patients with stage III disease treated with CMT.

**Figure 4 cancers-15-02213-f004:**
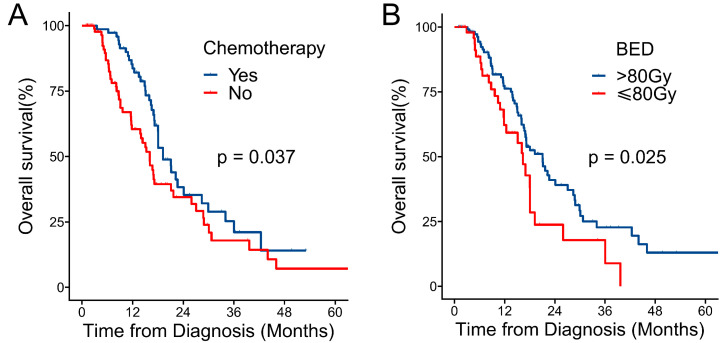
Kaplan–Meier plots of overall survival (OS) with chemotherapy (**A**) and BED (**B**) in the univariate analysis.

**Figure 5 cancers-15-02213-f005:**
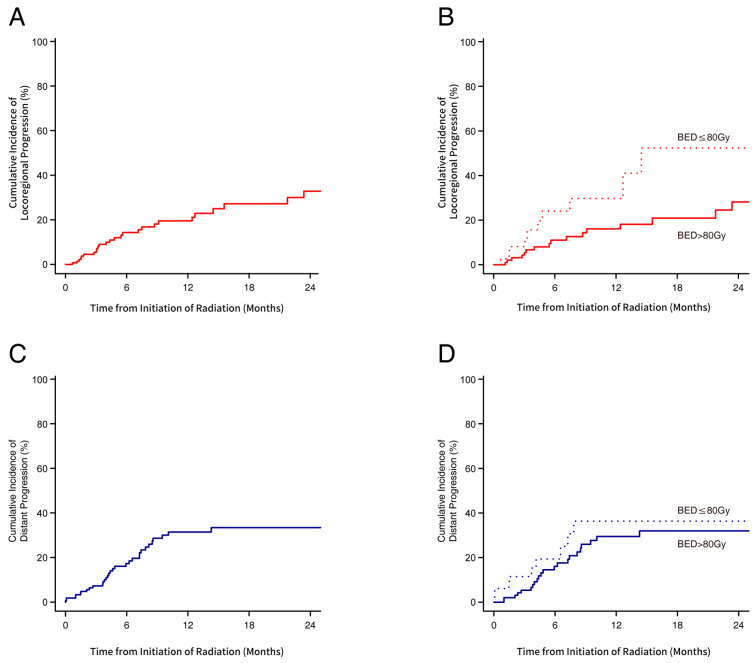
Cumulative incidence of disease progression. Cumulative incidence of locoregional progression (**A**); cumulative incidence of locoregional progression between BED dose groups (**B**); cumulative incidence of distant progression (**C**); cumulative incidence of distant progression between BED dose groups (**D**).

**Figure 6 cancers-15-02213-f006:**
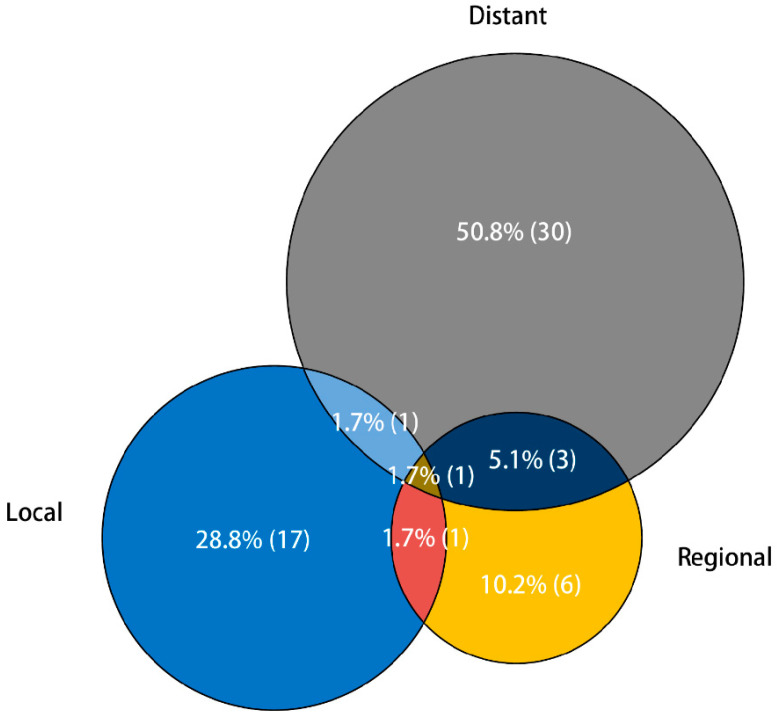
Patterns of failure.

**Table 1 cancers-15-02213-t001:** Demographic and Clinical Characteristics of Patients.

Characteristic	No. (%)
Median age at time of RT (range), year	64 (36–85)
Sex	
Female	69 (41.1)
Male	99 (58.9)
Tumor location	
Head	111 (66.1)
Body/tail	57 (33.9)
ECOG score	
0–1	92 (53.6)
2	76 (46.4)
Diagnosis confirmation	
Pathologically	39 (23.2)
Clinically	129 (76.8)
T Stage	
T1–T2	37 (22.0)
T3–T4	131(78.0)
N Stage	
N0	104 (61.9)
N+	64 (38.1)
AJCC Stage	
I	26 (15.5)
II	24 (14.3)
I–II	50 (29.8)
III	118 (70.2)
Tumor size, median (range), cm	3.8 (1.3–10.0)
Pre-RT CA19–9, median (range), U/mL	182.8 (0.6–20,000)
Chemotherapy	
No	95 (56.5)
Yes	73 (43.5)
Sequence of RT and chemotherapy	
CT-RT-CT	26 (35.6)
CT-RT	26 (35.6)
RT-CT	21 (28.8)
Pre-RT chemotherapy cycles	
≤6	42 (57.5)
>6	31 (42.5)
Total chemotherapy cycles	
≤8	43 (58.9)
>8	30 (41.1)
RT technique	
IMRT	31 (18.5)
SBRT	137 (81.5)
BED_10_	
≤80 Gy	54 (32.1)
>80 Gy	114 (67.9)

RT, radiotherapy; ECOG, Eastern Cooperative Oncology Group; AJCC, American Joint Committee on Cancer; CA19–9, carbohydrate antigen199; SBRT, stereotactic body radiotherapy; IMRT, intensity-modulated radiotherapy; BED, biologically effective dose; CT-RT-CT: induction chemotherapy followed by radiotherapy and consolidation chemotherapy; CT-RT: chemotherapy followed by radiotherapy; RT-CT: radiotherapy followed by chemotherapy.

**Table 2 cancers-15-02213-t002:** Univariate and multivariate analyses for overall survival from diagnosis.

Characteristics	mOS (Months)	Univariate Analysis		Multivariate Analysis	
		Hazard Ratio (95% CI)	*p* Value	Hazard Ratio (95% CI)	*p* Value
Age at time of RT	18.0	1.008 (0.991–1.025)	0.363		
Sex					
Female	17.1	Reference			
Male	18.0	0.935 (0.613–1.426)	0.755		
Tumor location					
Head	17.0	Reference			
Body/tail	21.1	0.853 (0.545–1.336)	0.488		
ECOG score					
0–1	18.0	Reference			
2	16.7	1.200 (0.793–1.816)	0.388		
Diagnosis confirmation					
Clinically	17.1	Reference			
Pathologically	18.0	0.797 (0.483–1.315)	0.375		
Pre-RT CA19–9					
≤130 U/mL	22.2	Reference			
>130 U/mL	16.1	1.623 (1.058–2.491)	0.027	1.765 (1.142–2.728)	0.011
Tumor size					
≤4 cm	21.7	Reference			
>4 cm	18.0	1.123 (0.737–1.709)	0.590		
T stage					
T1–T2	21.7	Reference			
T3–T4	17.6	1.276 (0.773–2.104)	0.340		
N stage					
N0	18.0	Reference			
N+	17.0	0.890 (0.579–1.369)	0.595		
AJCC Stage					
I–II	21.1	Reference			
III	17.0	1.525 (0.952–2.443)	0.079	1.726 (1.049–2.840)	0.032
Chemotherapy					
No	16.0	Reference			
Yes	19.2	0.642 (0.422–0.978)	0.039	0.509 (0.327–0.791)	0.003
RT technique					
SBRT	18.0	Reference			
IMRT	16.4	1.055 (0.627–1.774)	0.841		
BED_10_					
≤80 Gy	16.4	Reference			
>80 Gy	21.1	0.592 (0.373–0.938)	0.026	0.541 (0.332–0.881)	0.014

RT, radiotherapy; ECOG, Eastern Cooperative Oncology Group; AJCC, American Joint Committee on Cancer; CA19–9, carbohydrate antigen199; SBRT, stereotactic body radiotherapy; IMRT, intensity-modulated radiotherapy; BED_10_, biologically effective dose, α/β = 10.

**Table 3 cancers-15-02213-t003:** Treatment-related Toxicities.

Toxicity	Grade 1–2	Grade 3–4
Acute toxicity		
Nausea and vomiting	122 (72.6%)	11(6.5%)
Neutropenia	57 (33.9%)	5 (3.0%)
Thrombocytopenia	13 (7.7%)	3 (1.8%)
Hyperbilirubinemia	6 (3.6%)	1(0.6%)
Gastrointestinal ulcer	2 (1.2%)	0 (0.0%)
Late toxicity		
Gastric ulcer	5 (3.0%)	0 (0.0%)
Duodenal ulcer	14 (8.3%)	0 (0.0%)
Gastrointestinal obstruction	12 (7.1%)	4 (2.4%)

## Data Availability

The material contains the original contributions presented in our study. For further information, contact the corresponding author.

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
