# Peer review of "Survival Outcomes and Failure Patterns in Patients with Inoperable Non-Metastatic Pancreatic Cancer Treated with Definitive Radiotherapy"

_cancers, 2023, doi:10.3390/cancers15082213_

Round 1

Reviewer 1 Report

The paper is of interest given the numbers of patients invovled

However despite having read the paper several times I find the layout of the data almost incomprehensible.

There is such a variety of patients getting varying chemotherapy (including none) and RT regimes that making any comparison is v difficult.

Step 1 would be for the authors to decide how they want to present the data more clearly. Thereafter I would suggest a CONSORT diagram highlighting induction chemo vs post RT etc and separating hypofractionated RT vs long course

I would be happy to reconsider the paper once tis has been done

Reviewer 2 Report

Study is well performed, and roughly suggests that radiotherapy provides good locoregional control and possibly survival benefit in these groups of patients. Multivariate analysis suggests that the addition of chemotherapy and a higher BED improves outcome.

I think the authors go too far in making sub-subset analyses and drawing conclusions from those. They should leave those out.  Furthermore there are some issues concerning their local control analysis (enough data? Intention to treat?). Ant the discussion is very long and should in my view focus on the main finding: good local control and possibly improved survival.

Please find detailed comments below in document:  "remarks"

Round 2

Reviewer 1 Report

thank you for redrafting the paper. it reads far better now and I feel is appropriate for publication